# Diagnostic and Prognostic Nomograms for Hepatocellular Carcinoma Based on PIVKA-II and Serum Biomarkers

**DOI:** 10.3390/diagnostics13081442

**Published:** 2023-04-17

**Authors:** Shu An, Xiaoxia Zhan, Min Liu, Laisheng Li, Jian Wu

**Affiliations:** 1Department of Laboratory Medicine, The First Affiliated Hospital, Sun Yat-sen University, Guangzhou 510080, China; 2Center of Hepato-Pancreato-Biliary Surgery, The First Affiliated Hospital, Sun Yat-sen University, Guangzhou 510080, China

**Keywords:** hepatocellular carcinoma, PIVKA-II, AFP, nomogram, diagnosis, prognosis

## Abstract

Background: The aim of the present study was to develop an improved diagnostic and prognostic model for HBV-associated HCC by combining AFP with PIVKA-II and other potential serum/plasma protein biomarkers. Methods: A total of 578 patients, including 352 patients with HBV-related HCC, 102 patients with HBV-associated liver cirrhosis (LC), 124 patients with chronic HBV, and 127 healthy subjects (HS), were enrolled in the study. The serum levels of AFP, PIVKA-II, and other laboratory parameters were collected. Univariate and multivariate logistic regression and Cox regression analyses were performed to identify independent diagnostic and prognostic factors, respectively. The diagnostic efficacy of the nomogram was evaluated using receiver operator curve (ROC) analysis and the prognostic performance was measured by Harrell’s concordance index (C-index). Results: AFP and PIVKA-II levels were significantly increased in HBV-related HCC, compared with those in HBV-associated LC and chronic HBV participants (*p* < 0.05 and *p* < 0.001, respectively). The diagnostic nomogram, which included age, gender, AFP, PIVKA-II, prothrombin time (PT), and total protein (TP), discriminated patients with HBV-HCC from those with HBV-LC or chronic HBV with an AUC of 0.970. In addition, based on the univariate and multivariate Cox regression analysis, PIVKA-II, γ-glutamyl transpeptidase, and albumin were found to be significantly associated with the prognosis of HBV-related HCC and were incorporated into a nomogram. The C-index of the nomogram for predicting 3-year survival in the training and validation groups was 0.75 and 0.78, respectively. The calibration curves for the probability of 3-year OS showed good agreement between the nomogram prediction and the actual observation in the training and the validation groups. Furthermore, the nomogram had a higher C-index (0.74) than that of the Child−Pugh grade (0.62), the albumin−bilirubin (ALBI) score (0.64), and Barcelona Clinic Liver Cancer (0.56) in all follow-up cases. Conclusion: Our study suggests that the nomograms based on AFP, PIVKA-II, and potential serum protein biomarkers showed a better performance in the diagnosis and prognosis of HCC, which may help to guide therapeutic strategies and assess the prognosis of HCC.

## 1. Introduction

Hepatocellular carcinoma (HCC) is one of the most prevalent cancers and one of the leading causes of cancer-related death globally, with an estimated 905,677 new cases and 830,180 new deaths in 2020 [1]. The dominant histologic type of primary liver cancer is hepatocellular carcinoma, accounting for nearly 75–90% of cases [2]. The majority of cases of liver cirrhosis and hepatocellular carcinoma (HCC) are attributed to persistent infections with the hepatitis B virus (HBV) and hepatitis C virus (HCV) [3]. Moreover, the high number of chronic HBV carriers are responsible for the high prevalence of HCC in China [4]. Unfortunately, lack of surveillance and inadequate early diagnosis account for the poor prognosis and high mortality of HCC [5,6]. Therefore, the accurate and effective diagnosis of HCC is essential to rapidly determine potentially curative therapies, such as liver resection or transplantation [6,7].

Currently, abdominal ultrasound with serum alpha-fetoprotein (AFP) assessment is recommended for the early detection of HCC according to the guidelines [6,8]. Significant heterogeneity in tumor size, equipment, and research experience compromises the efficiency of abdominal ultrasound in HCC detection [9,10]. In addition, HBV and cirrhosis patients with serum AFP levels up to 20 ng/mL at 12 months after entecavir treatment are most likely to develop into HCC [11,12]. However, approximately 30% of patients with liver cancer are always negative for serum AFP [13]. AFP detection has a sensitivity of only 46% to 59% for clinical HCC diagnosis and only 40% for preclinical prediction [14,15]. Therefore, the sensitivity and specificity of these methods in the screening and diagnosis of small and early HCC remain inefficient [16]. There is an urgent need to explore new strategies for the identification of high-risk groups for liver cancer and for the screening of HCC patients, especially those with early-stage HCC, AFP-negative HCC, and micro-liver cancer in the subclinical stage [6].

Protein-induced vitamin K absence or antagonist II (PIVKA-II) is widely recognized as a reliable biomarker for the diagnosis, prognosis, treatment response, and recurrence monitoring of HCC [17,18,19]. Studies have demonstrated that PIVKA-II levels of 40 mAU/mL are indicative of early HCC with sensitivity and specificity rates of 64% and 89%, respectively [20]. Furthermore, PIVKA-II shows a significant diagnostic value for AFP-negative liver cancer [21,22]. Although PIVKA-II is noted as a highly specific biomarker for HCC, the limited sensitivity of PIVKA-II raises concerns about its efficacy as a surveillance biomarker, as alcohol-related liver disease, obstructive jaundice, and cholestasis can also influence the expression of PIVKA-II [23,24]. Therefore, combination studies of AFP and PIVKA-II have been recommended to improve the sensitivity and specificity of HCC screening [15].

Several serum biomarkers have been shown to assist in the diagnosis and prognosis of HCC, including liver enzyme indicators (aspartate aminotransferase (AST), alanine aminotransferase (ALT), alkaline phosphatase (ALKP), and γ-glutamyl transpeptidase (GGT)), or indicators reflecting liver metabolism (total bilirubin (TBIL)) and protein synthesis function (total protein (TP), albumin (ALB), and prothrombin time (PT)) [25,26,27,28,29,30]. To fully explore and harness the potential benefits of different biomarkers, it is necessary to identify potential serum/plasma protein biomarkers that can be combined with AFP and PIVKA-II to facilitate HCC detection and surveillance. The present study aimed at optimizing AFP and PIVKA-II related diagnostic and prognostic models by integrating potential serum/plasma protein biomarkers in HCC.

## 2. Materials and Methods

### 2.1. Data Collection

From January 2014 to December 2016, individuals with HBV infection, HBV-related liver cirrhosis (HBV-LC), and HBV-HCC, as well as healthy subjects (HS) from the First Affiliated Hospital of Sun Yat-sen University (FAH-SYSU), were enrolled in the training cohort. From June 2017 to December 2018, individuals with HBV, HBV-LC, HBV-HCC, and HS from FAH-SYSU were recruited for the validation cohort. The exclusion criteria were as follows: (a) history of previous treatment (hepatic resection, liver transplant, trans-arterial chemoembolization, radiofrequency, anti-angiogenetic drugs, and warfarin therapy); (b) Child−Pugh C; (c) obstructive jaundice; (d) estimated creatinine clearance <30 mL/min; (e) diagnosis of second extrahepatic neoplasia; and (f) metastasis. HBV infection was defined as hepatitis B surface antigen (HBsAg) positivity within the previous 6 months. HBV-HCC was defined as HCC with HBV infection, excluding alcoholic liver disease or hepatitis C virus infection [6]. Patients with HCC met the diagnostic criteria for HCC, including imaging evidence (ultrasound, computed tomography, and MRI) and histopathological confirmation [6]. Liver cirrhosis was diagnosed based on clinical parameters, including histologic examination, laboratory tests, and radiologic or endoscopic evidence of cirrhosis [31]. HS were blood donors with no history of chronic liver disease or gastrointestinal malignancy. The study was approved by the Institutional Review Boards at FAH-SYSU and informed consent was obtained from each participant.

### 2.2. Laboratory Methods

Peripheral blood was collected from each participant and centrifuged at 800× *g* for 10 min. The serum was aliquoted and immediately frozen at −80 °C until testing. Serum concentrations of AFP, PIVKA-II, carcinoembryonic antigen (CEA), and carbohydrate antigen 199 (CA199) were determined using the ARCHITECT immunoassay according to protocol (Abbott Diagnostics). Clinical laboratory test results, including biochemical indices, blood routine results, and coagulation function results, were collected from routine clinical practice. The clinical laboratory test results included glutamic-oxalacetic transaminase (AST), glutamic-pyruvic transaminase (ALT), albumin (ALB), total protein (TP), lactate dehydrogenase (LDH), total bilirubin (TBIL), direct bilirubin (DBIL), γ-glutamyl transpeptidase (GGT), and HBV surface antigen (HBsAg). Blood routine indexes included white blood cell (WBC), neutrophil (NET), lymphocyte (LY), neutrophil-to-lymphocyte ratio (NLR), red blood cell (RBC), red cell distribution width (RDW), hemoglobin (Hb), platelet count (PLT), and mean platelet volume (MPV). Indicators of the coagulation function included prothrombin time (PT), thrombin time (TT), activated partial thromboplastin time (APTT), fibrinogen (FIB), and international normalized ratio (INR). All of the biochemical indices, blood routine indexes, and coagulation function results were acquired via standard automated laboratory methods and utilizing commercially available kits according to the manufacturer’s protocols.

### 2.3. Statistical Analysis

For demographic data, categorical variables were expressed as a mean ± standard deviation (SD), range, or ratio. Univariate analysis was performed using the Mann−Whitney U-test, chi-squared test, or Fisher’s exact test, as appropriate. Variables with right-skewed distributions, including AFP and PVIKA-II, were log-transformed before logistic or Cox regression analysis. OS was defined as the interval from the date of treatment to the date of patient death or lost follow-up. Univariate and multivariate logistic regression and Cox regression analyses were performed to identify independent diagnostic and prognostic factors, respectively. Variables with *p* < 0.05 were included in the multivariate regression analyses. The nomogram was constructed based on the results of the multivariate analysis utilizing the rms package. The discriminative performance of the diagnostic nomogram model was examined using area under the curve (AUC) analysis [32]. In addition, the final variables for the construction of the prognostic nomogram were selected using the backward step-down method based on the Akaike information criterion (AIC) [33]. To evaluate the discriminative ability of the prognostic nomogram, Harrell’s concordance index (C-index) and receiver operating characteristic (ROC) curve were used. Calibration curves were plotted to evaluate the predictive accuracy of the prognostic nomogram. Furthermore, the discriminatory ability of the prognostic nomograms was compared with the HCC prognostic model of Barcelona Clinic Liver Cancer, ALBI score, and Child−Pugh grade by analyzing the ROC curves. The results were considered statistically significant if the *p* value was less than 0.05. Statistical analysis was accomplished using the SPSS software version 25.0 and R (version 3.4.2).

## 3. Results

### 3.1. Characteristics of Patients

A total of 705 subjects were included in the study. The clinical laboratory characteristics of the patients with HBV-related HCC, HBV-associate liver cirrhosis (LC), chronic HBV infection, and healthy subjects are shown in Table 1. Consistent with previous studies, the serum concentrations of liver function biomarkers AST and ALT were significantly higher in patients with HCC than in other patients (*p* < 0.001). Serum levels of ALB, TP, PLT, and Hb were lower in the HCC and cirrhosis groups than in the hepatitis and normal control groups. In addition, TBIL was increased moderately in the HCC group, but was significantly elevated in the LC group. The time of PT was the longest in the cirrhosis group, and PLT and Hb were dramatically decreased in the cirrhosis group.

### 3.2. AFP and PIVKA-II Distribution among Disease Groups

As depicted in Figure 1, there were significant differences between the HCC group and the other groups in the serum concentration of PIVKA-II and AFP (*p* < 0.001 and *p* < 0.05, respectively). Among 352 patients with HCC, 125 (35.51%) were AFP negative when the positive screening value of AFP was defined as >20 ng/mL [34]. Similarly, when the positive screening value of PIVKA-II was defined as >40 mAU/mL [35], 84.37% of patients with HCC had increased serum levels of PIVKA-II. Notably, elevated levels of PIVKA-II were observed in 100 (80.0%) AFP negative patients. Furthermore, the sensitivity for AFP in combination with PIVKA-II to predict HCC was 92.89% (Appendix A). These results suggest that PIVKA-II is a complementary biomarker to AFP in differentiating HBV-HCC.

### 3.3. Diagnostic Factors of HCC Based on Logistic Regression

The results based on the univariate and multivariate logistic regression analysis to assess the predictive value of biomarkers for HCC are shown in Table 2. The results of the univariate regression analysis indicate that night variables, including age, gender, AFP, PIVKA-II, AST, ALT, ALB, PT, and TP, were significant predictors of HCC when compared with HBV infection or liver cirrhosis. In the multivariate logistic regression analysis, increasing age (OR: 1.08, 95% CI 1.050–1.116; *p* < 0.001), male gender (OR: 2.662, 95% CI 1.159–6.117; *p* = 0.021), elevated AFP (OR: 8.291, 95% CI 4.366–15.743; *p* < 0.001), increased PIVKA-II (OR: 12.231, 95% CI 5.853–25.559; *p* < 0.001), decreased TP (OR: 0.912, 95% CI 0.871–0.954; *p* < 0.001), and prolonged PT (OR: 0.839, 95% CI 0.752–0.935; *p* = 0.002) were independent predictors of HCC.

Based on the results of the multivariate regression analysis, risk factors including age, gender, AFP, PIVKA-II, PT, and TP were incorporated into a nomogram, referred to as the APPT grade. According to the nomogram, log-transformed AFP and PIVKA-II values had the most significant impact on early HCC diagnosis, followed by PT, TP, age, and sex. The discriminative power of the nomogram was evaluated using ROC curves and the AUC of the diagnostic nomogram was 0.970 (Figure 2). The ROC analysis indicated that the diagnostic nomogram demonstrated an excellent ability to discriminate HCC from CHB and LC.

### 3.4. Clinical Characteristics of HCC Patients Enrolled in Survival Analysis

A total of 241 HCC patients were successfully followed up for 3 years. Correspondingly, in our data, 106 patients (44%) died during the 3-year follow-up. Non-survivors had significantly lower levels of ALB and higher levels of PIVKA-II, CA199, RDW, AST, GGT, TBIL, DBIL, PT, and FIB (Table 3, *p* < 0.05).

From January 2014 to December 2016, 142 HCC patients were successfully followed up and enrolled in the training group, while from June 2017 to December 2018, 99 HCC patients were followed up and enrolled in the validation group. The characteristics and laboratory indexes of the HCC patients in the training and validation cohorts are shown in Appendix A. Univariable and multivariable Cox regression analyses were applied to identify the independent prognostic factors for HCC patients in the training cohort (Table 4). Univariate Cox regression analyses revealed that PIVKA-II, GGT, DBIL, FIB, and ALB were significant predictors of survival in HCC patients. Variables with a *p* value of <0.05 in univariable analysis were included in the multivariable analysis. The results of multivariate Cox regression analysis demonstrated that log_10_PIVKA-II (HR: 1.347, 95% CI: 1.121–1.780), GGT (HR: 1.002, 95% CI: 1.001–1.003), and ALB (HR: 0.932, 95% CI: 0.888–0.979) were independent variables for the prognosis of HCC (Table 4).

### 3.5. Predictive Potentials of Prognostic Nomogram

The prognostic nomogram was established based on the risk factors PIVKA-II, GGT, and ALB, which were identified by the multivariate analysis. Higher levels of PIVKA-II and GGT and lower levels of ALB during hospitalization were associated with a poorer prognosis for HCC patients. The nomogram based on PIVKA-II, GGT, and ALB was referred to as the PGA grade (Figure 3).

The discriminative power of the nomogram was assessed using Harrell’s concordance index and ROC curves. The C-indexes for the prediction of overall survive in the training and validation groups were 0.75 (95% CI 0.67–0.83) and 0.78 (95% CI 0.68–0.88), respectively (Figure 4). Furthermore, the calibration curves for the probability of 3-year OS demonstrated good agreement between prediction by the nomograms and the actual observation in the training and the validation sets (Figure 4). Therefore, the PGA grade exhibited a great prediction efficiency for the 3-year prognosis of HCC patients.

Child−Pugh grade, ALBI score, and BCLC were utilized to predict the prognosis of HCC. To further evaluate the clinical value of the PGA grade, we compared the performance of the predictive potential of 3-year survival probability in Child−Pugh grade, ALBI score, BCLC, and PGA grade in all follow-up cases. The C-index for OS prediction of the PGA grade was 0.74 (95% CI 0.68–0.80), which was found to be superior to that of the Child−Pugh grade (0.62), ALBI score (0.64), and BCLC (0.56). The result suggests that the PGA grade has better predictive potential than the ALBI score, Child−Pugh grade, and BCLC (Figure 5).

## 4. Discussion

The high incidence rate and mortality of HCC patients remains a major clinical concern. Early intervention based on risk stratification is an effective strategy to improve the survival rate of HCC patients. Clinical and histopathological parameters, including tumor burden, vascular invasion, lymph node, and extrahepatic metastasis, are closely related to the prognosis of HCC [6,36]. As accurate histological diagnosis is not easily available, and ultrasonography is far from excellent, serum biomarker assessments are more objective and accurate at predicting and evaluating the prognosis of HCC [10,14]. Herein, we presented a reliable and easy-to-use model for better diagnosis or prognosis of early HCC by integrating potential serum biomarkers. The diagnostic nomogram based on age, gender, AFP, PIVKA, PT, and TP (refer to APPT grade) can improve the diagnostic efficacy of early HCC. In addition, the prognostic nomogram, constructed by PIVKA, GGT, and ALB (refer to PGA grade) exhibited excellent discrimination between survival and non-survival HCC patients. Moreover, the prognostic nomogram (PGA grade) outperformed the routinely used prognostic models including ALBI grade, BCLC, and CTP classification. A high PGA grade may serve as an effective predictor of survival rate in HCC and support optimal therapeutic selection in HCC.

Collective evidence suggests a positive correlation between AFP and PIVKA-II levels and clinicopathological performance, such as tumor size, tumor differentiation, and vascular invasion [18,21,37]. In the present study, using 20 ng/mL as a cut-off, 35.51% of HCC patients were AFP negative, which is consistent with previous reports indicating that approximately 30% of liver cancer patients were consistently negative for serum AFP [13]. The sensitivity of AFP for HCC in CHB patients was 64.49%. On the other hand, PIVKA-II with a level of up to 40 ng/mL revealed a superior specificity of 84.37% for HCC diagnosis. Furthermore, the combination of AFP and PIVKA-II further increased the specificity to 92.89%. Our results were consistent with the conclusions of other researchers who suggested that the combination of AFP and PIVKA-II can improve the diagnosis of HBV-related HCC [22,37].

The occurrence and development of HCC involves a multi-step evolutionary process from the molecular to the clinical level, characterized by marked abnormalities in liver function, including liver enzymes, metabolism, and protein synthesis function [38]. Emerging evidence suggests that the imbalance of tumor and blood coagulation disorders promotes tumor growth, invasion, and metastasis [39,40]. For example, prolonged prothrombin time (PT) has been associated with aggressive tumor growth and poor survival rates in various cancers, such as lung cancer [39], kidney cancer [41], and early HCC [40]. After performing univariate and multivariate logistic regression analyses, we identified PT as one of the independent risk factors for HCC (OR = 0.839, *p* = 0.002). In addition, hypoproteinemia, especially hypoalbuminemia, suggests poor nutritional status and decreased hepatic synthesis function due to chronic liver disease and HCC [27,30]. According to our study, TP was also an independent risk factor for HCC (OR = 0.912, *p* < 0.001).

Therefore, the objective of this study was to construct a reliable and precise nomogram for the prediction of HCC by integrating biomarkers such as AFP, PIVKA-II, PT, and TP. The AUROC of the diagnostic nomogram was 0.970, indicating the critical role of APPT grade in the early detection of HBV-related HCC. By evaluating the individualized potential to develop HCC, the APPT grade may enable physicians to optimize the implementation and efficiency of a screening surveillance strategy.

Liver function indicators, including GGT, ALB, PT, and TP, are critical serum biomarkers for determining hepatic reserve function, which is essential for the prognosis of HCC. Currently, several prognostic models, such as BCLC [42], albumin−bilirubin (ALBI) grade [43,44], and Child−Turcotte−Pugh (CTP) classification, have been established and validated as effective tools for predicting HCC outcomes [44,45,46]. Despite the widespread use of CTP scores to evaluate preoperative liver function, the limitations regarding its subjective variables, such as clinical grading of ascites and encephalopathy, have been extensively discussed [6,47]. The albumin−bilirubin (ALBI) score has the specific advantage of being based on statistical and objective evidence [44], but our multivariable Cox analyses showed no correlation between bilirubin levels and worse survival in HCC patients. Although previous studies have reported that HCC patients with higher bilirubin levels had a worse prognosis, our results suggested otherwise.

PIVKA-II is a novel prognostic predictor for HCC as elevated levels are associated with early recurrence, vascular invasion, large HCC size, and poor prognosis [17,22]. However, the elevation of PIVKA-II is not specific to HCC because interfering factors such as taking warfarin, primary gastric adenocarcinoma, vitamin K deficiency, inflammatory bowel disease, intestinal flora imbalance, renal failure, malnutrition, and alcoholic liver disease can lead to elevated serum levels of PIVKA-II in non-HCC patients [48]. In the current work, the prognostic value of PIVKA-II in combination with other liver function parameters, especially GGT and ALB, was evaluated through survival analysis in HCC patients. High levels of GGT protein, which resulted from impaired biliary excretion in gastrointestinal cancer or the secretion of HCC cells, were positively associated with a large tumor size and advanced TNM stage, and were considered as an independent prognostic factor for predicting the survival rate of individuals with AFP-negative HCC [49,50,51]. Consistent with previous studies, the results of our analysis suggest that the increased GGT level seemed to be a strong risk factor for an unfavorable survival rate in patients with HCC. Although PIVKA-II, GGT, and ALB partially reflect different aspects of HCC including tumor burden, vascular invasion, and poor tumor differentiation, respectively, they may complement each other when used in combination. Therefore, the combined use of PIVKA-II, GGT, and ALB could validate hepatic reserve function and increase their predictive probability in the prognosis of HCC. In the present study, when PIVKA-II was used in combination with GGT and ALB (PGA grade), the PGA grade (C-index: 0.74) exhibited an excellent discrimination and good accuracy and outperformed commonly used prognostic models, including ALBI grade (0.64), BCLC (0.56), and CTP classification (0.62).

Furthermore, Xu and colleagues conducted a study to evaluate the predictive efficacy of GGT for prognosis in patients with HCC who underwent liver resection [52]. The study found that elevated GGT levels were significantly associated with a higher risk in this patient population, with an AUC of 0.643. In another study, Park et al. [53] investigated the clinical utility of the response of AFP and PIVKA-II on the prognosis of patients with locally advanced HCC who received local treatment. The combination of AFP and PIVKA-II had a prognostic power of 0.626 for overall survival, which was better than AFP alone (0.592). Although the management of patients may vary among different studies, encompassing variations in therapeutic schedules and etiology, our results suggest that the proposed PGA grade could improve prognostic prediction compared with using GGT alone or combining AFP and PIVKA. In addition, several nomograms have been developed to predict recurrence and survival in HCC patients who had undergone resection treatment [54,55,56]. Wang et al. reported that preoperative TACE therapy, microvascular invasion (MVI), AFP, ALBI grade, tumor differentiation, tumor size, intraoperative blood transfusion, and surgical modality were independent risk factors for overall survival in patients with single large and huge HCC who underwent curative resection treatment [54]. Based on these risk factors, a nomogram was developed that achieved high C-indexes (0.86 for overall survival). Compared with these nomograms [54,55,56], which are based on clinical data such as tumor size and vascular invasion, the PGA grade was found to have less potential for predicting prognosis, possibly due to the lack of clinical characteristics, different treatment methods received by patients and the different histopathological types of liver cancer.

The present study has some limitations that need to be considered. First, the lack of clinical characteristics, such as treatment methods, tumor size, tumor differentiation, and vascular invasion, may have led to confounding deviations. Therefore, further prospective research is warranted, which should include a comprehensive evaluation of both clinical and laboratory data to devise a better method for the diagnosis and prognosis of HCC. Second, this study was conducted at a single center and had a relatively small sample size. Therefore, to draw a precise conclusion, our results need to be validated in a larger, multicenter clinical trial, which should include long-term follow-up evaluations.

In conclusion, the combination of AFP and PIVKA with PT and TP showed the greatest diagnostic ability for HBV-related HCC, while the combination of PIVKA-II with GGT and ALB was efficient at validating the hepatic reserve function and predicting the prognosis of HCC. Therefore, clinicians should select the most appropriate biomarkers for HCC, which can facilitate better assessment and guide appropriate therapeutic strategies for HCC.

## Figures and Tables

**Figure 1 diagnostics-13-01442-f001:**
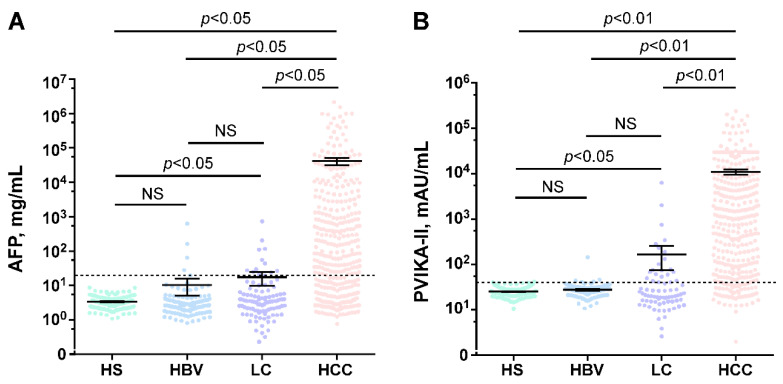
Serum AFP and PIVKA-II concentrations in different populations. (**A**) Serum AFP levels in healthy subjects (HS), chronic HBV infections (HBV), HBV-associate liver cirrhosis (LC), and patients with HBV-related HCC. (**B**) Serum PIVKA-II levels in healthy subjects (HS), chronic HBV infection (HBV), HBV-associated liver cirrhosis (LC), and patients with HBV-related HCC. AFP reference line: 20 ng/mL and PIVKA-II reference line: 40 mAU/mL.

**Figure 2 diagnostics-13-01442-f002:**
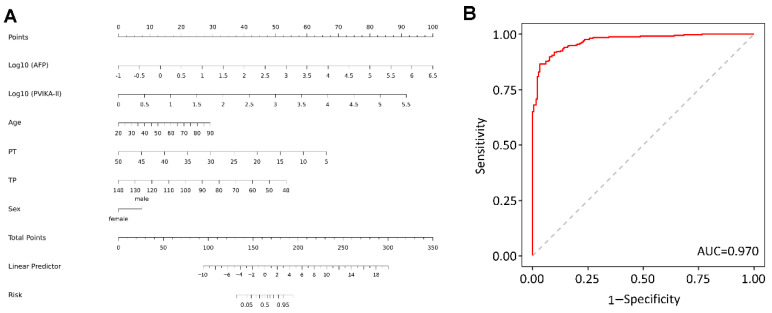
Diagnostic nomogram for patients with HCC. (**A**) A diagnostic nomogram for differentiating HCC cases from chronic HBV and LC. (**B**) AUROC for the diagnostic nomogram in differentiating HCC cases from chronic HBV and LC.

**Figure 3 diagnostics-13-01442-f003:**
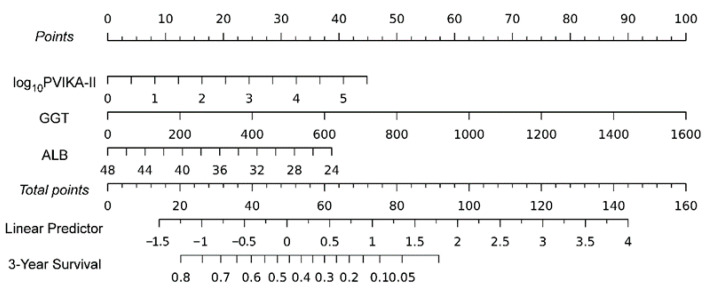
A prognostic nomogram for the survival at the 3-year follow-up in hepatocellular carcinoma.

**Figure 4 diagnostics-13-01442-f004:**
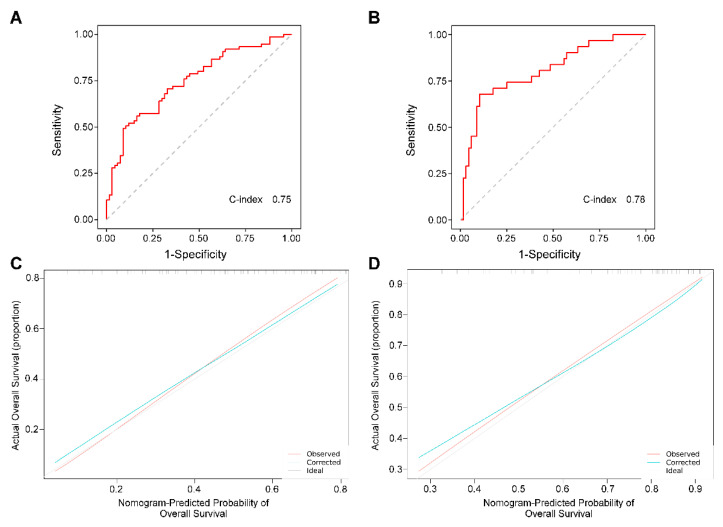
The C-index of the nomogram for the prognostic prediction of OS in the training cohort (**A**) and validation cohort (**B**). The calibration curves of OS based on nomogram prediction and the actual observation in the training cohort (**C**) and validation cohort (**D**).

**Figure 5 diagnostics-13-01442-f005:**
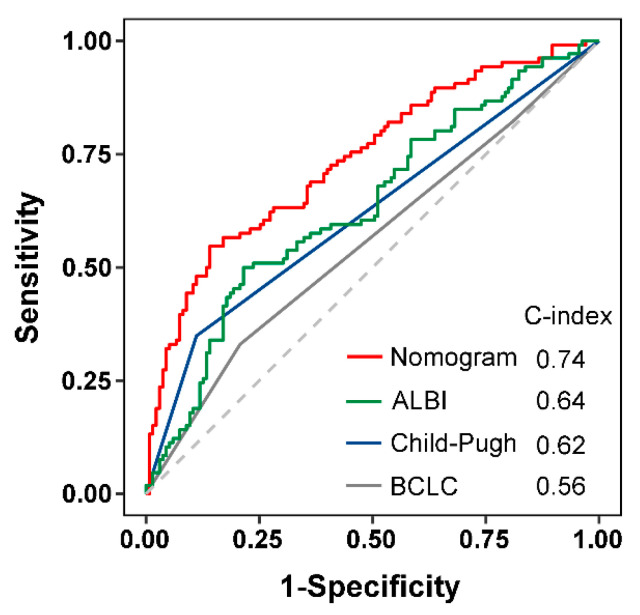
Comparison of predictive accuracy between ALBI score, Child−Pugh grade, BCLC, and the PGA grade. BCLC, Barcelona Clinic Liver Cancer staging; ALBI, albumin−bilirubin score; Child−Pugh, Child−Pugh grade.

**Table 1 diagnostics-13-01442-t001:** Clinical characteristics of the study population.

Parameters	HCC (N = 352)	Liver Cirrhosis (N = 102)	HBV (N = 124)	Healthy Subjects (N = 127)
Age, years mean (SD)	54 (11.8)	55.7 (12.7)	36.3 (10.8)	50.1 (10.7)
Gender, n (%)				
Male	309 (87.8%)	67 (65.7%)	83 (66.9%)	69 (54.3%)
Female	43 (12.2%)	35 (34.3%)	41 (33.1%)	58 (45.7%)
AST (U/L) mean (SD)	92.2 (135.2)	79.2 (91.9)	40.0 (59.5)	24.1 (6.1)
ALT (U/L) mean (SD)	79.4 (190.2)	52.1 (56.5)	45.5 (120.8)	24.0 (13.6)
TBIL, μmol/L mean (SD)	32.5 (57.0)	60.7 (96.2)	17.0 (14.5)	12.4 (3.9)
ALB (g/L) mean (SD)	36.3 (5.4)	32.7 (6.6)	45.6 (3.0)	44.5 (3.8)
TP (g/L) mean (SD)	66.5 (8.5)	66.2 (9.0)	75.4 (4.7)	73.1 (4.8)
PLT (×10^9^/L) mean (SD)	174.3 (90.3)	124.2 (87.8)	214.5 (47.9)	244.6 (53.8)
Hb (g/L) mean (SD)	129.5 (23.6)	106.7 (27.4)	145.2 (16.7)	146.1 (14.8)
PT, s mean (SD)	13.2 (2.8)	15.7 (4.5)	12.8 (2.0)	13.0 (1.1)

Categorical and continuous variables are presented as frequencies or mean (SD), respectively. AST: aspartate aminotransferase; ALT: alanine aminotransferase; TBIL: total bilirubin; ALB: albumin; TP: total protein; PLT: platelets; Hb: hemoglobin; PT: prothrombin time.

**Table 2 diagnostics-13-01442-t002:** Univariate and multivariate logistic regression analysis between HCC and liver cirrhosis or HBV infection groups.

Parameters	Univariate Analysis		Multivariate Analysis	
	OR (95% CI)	*p* Value	OR (95% CI)	*p* Value
Age	1.052 (1.037–1.066)	<0.001	1.082 (1.05–1.116)	<0.001
Gender (male)	3.641 (2.388–5.551)	<0.001	2.662 (1.159–6.117)	0.021
Log_10_AFP	6.572 (4.515–9.566)	<0.001	8.291 (4.366–15.743)	<0.001
Log_10_PVIKA-II	17.056 (9.366–31.059)	<0.001	12.231 (5.853–25.559)	<0.001
AST	1.004 (1.002–1.007)	0.001	0.996 (0.989–1.004)	0.339
ALT	1.004 (1.001–1.006)	0.01	1.009 (0.999–1.019)	0.078
TBIL	0.999 (0.996–1.002)	0.426		
ALB	0.923 (0.899–0.948)	<0.001	1.073 (0.998–1.153)	0.056
TP	0.933 (0.912–0.954)	<0.001	0.912 (0.871–0.954)	<0.001
PLT	1.00 (0.998–1.002)	0.968		
Hb	1.002 (0.996–1.009)	0.487		
PT	0.918 (0.862–0.977)	0.007	0.839 (0.752–0.935)	0.002

**Table 3 diagnostics-13-01442-t003:** Characteristics of HCC patients according to 3-year mortality.

Parameters	Survivor (n = 135)	Non-Survivor (n = 106)	*p* Value
Age (years) Median (IQR)	55.0 (48.0–63.0)	54.5 (48–65.0)	0.730 ^a^
Gender Male (%)	117 (86.7%)	93 (87.7%)	0.806 ^b^
Child-Pugh grade Number (%)			<0.001 ^b^
A	120 (88.9%)	69 (65.1%)	
B	15 (11.1%)	37 (34.9%)	
BCLC stage Number (%)			0.092 ^b^
A	26 (19.3%)	19 (17.9%)	
B	81 (60.0%)	52 (49.1%)	
C	28 (20.7%)	35 (33.0%)	
HBsAg Positive (%)	108 (80.0%)	86 (81.1%)	0.826 ^b^
AFP (mg/mL) Median (IQR)	49.8 (8.3–719.0)	92.9 (8.5–8406.2)	0.099 ^a^
PVIKA-II (mAU/mL) Median (IQR)	296.6 (61.0–1725.3)	1626.3 (123.2–10833.8)	<0.001 ^a^
CEA (mg/mL) Median (IQR)	2.8 (1.5–4.2)	2.6 (1.7–4.2)	0.948 ^a^
CA199 (mg/mL) Median (IQR)	7.4 (4.1–14.0)	12.5 (5.6–31.8)	0.001 ^a^
WBC (×10^9^/L) Median (IQR)	7.2 (5.1–10.2)	7.7 (5.5–9.5)	0.858 ^a^
LY (×10^9^/L) Median (IQR)	1.1 (0.8–1.6)	1.0 (0.7–1.5)	0.240 ^a^
NET (×10^9^/L) Median (IQR)	5.0 (3.4–7.9)	5.0 (3.1–7.8)	0.918 ^a^
NLR Median (IQR)	4.5 (2.7–8.4)	4.6 (2.7–8.9)	0.577 ^a^
RBC (×10^9^/L) Median (IQR)	4.0 (3.4–4.4)	4.1 (3.4–4.5)	0.230 ^a^
Hb (g/L) Median (IQR)	131.0 (117.0–145.5)	130.0 (113.3–142.0)	0.460 ^a^
RDW (%) Median (IQR)	13.5 (13.0–14.0)	14.0 (13.0–15.2)	0.049 ^a^
PLT (×10^9^/L) Median (IQR)	166.0 (113.0–223.0)	163.5 (99.5–243.3)	0.913 ^a^
MPV (fL) Median (IQR)	10.0 (9.2–11.1)	10.0 (9.2–10.8)	0.686 ^a^
ALT (U/L) Median (IQR)	39.0 (24.0–66.5)	45.0 (32.0–65.8)	0.109 ^a^
AST (U/L) Median (IQR)	39.0 (29.0–64.0)	56.5 (37.5–98.5)	<0.001 ^a^
GGT Median (IQR)	64.5 (39.2–112.7)	138.5 (72.7–242.7)	<0.001 ^a^
LDH Median (IQR)	228.5 (195.5–298.0)	250.5 (206.2–345.5)	0.072 ^a^
TBIL (µmol/L) Median (IQR)	16.4 (12.5–24.8)	19.7 (14.1–34.9)	0.004 ^a^
DBIL (µmol/L) Median (IQR)	3.8 (2.5–6.8)	5.5 (3.2–12.9)	0.002 ^a^
ALB (g/L) Median (IQR)	37.0 (34.8–40.7)	35.2 (31.7–39.0)	0.002 ^a^
TP (g/L) Median (IQR)	66.2 (61.3–71.2)	66.9 (61.2–71.6)	0.630 ^a^
Cr (µmol/L) Median (IQR)	71.0 (60.0–81.0)	67.0 (56.0–79.0)	0.114 ^a^
PT (s) Median (IQR)	12.4 (11.8–13.1)	12.8 (12.1–14.2)	0.004 ^a^
INR Median (IQR)	1.1 (1.1–1.2)	1.1 (1.0–1.2)	0.949 ^a^
APTT (s) Median (IQR)	29.2 (27.2–33.4)	30.8 (27.6–35.6)	0.066 ^a^
TT (s) Median (IQR)	18.3 (17.5–19.1)	17.9 (17.2–19.1)	0.316 ^a^
FIB (g/L) Median (IQR)	2.6 (2.1–3.4)	3.1 (2.1–4.0)	0.035 ^a^

Categorical and continuous variables are presented as frequencies or medians (IQR), respectively. Differences between groups in categorical and continuous variables are analyzed using the chi-squared test and Mann−Whitney U test, respectively. BCLC: Barcelona Clinic Liver Cancer; HBsAg: HBV surface antigen; CEA: carcinoembryonic antigen; CA199: carbohydrate antigen 199; LY: lymphocyte; NET: neutrophils; NLR: neutrophil-to-lymphocyte ratio; RDW: red cell distribution width; MPV: mean platelet volume; Cr: creatinine; TBIL: total bilirubin; DBIL: direct bilirubin; INR: international normalized ratio; APTT: activated partial thromboplastin time; TT: thrombin time; FIB: fibrinogen. ^a^ Wilcoxon rank sum test; ^b^ Chi-squared test.

**Table 4 diagnostics-13-01442-t004:** Univariate and multivariate Cox regression analysis in HCC patients according to 3-year mortality.

Parameters	Univariate Analysis		Multivariate Analysis	
	HR (95% CI)	*p* Value	HR (95% CI)	*p* Value
Log_10_PVIKA-II	1.573 (1.263–1.958)	<0.001	1.347 (1.053–1.724)	0.018
GGT	1.005 (1.002–1.008)	<0.001	1.002 (1.001–1.003)	0.002
CA199	1.004 (1.002–1.006)	<0.001	1.001 (0.998–1.004)	0.395
AST	1.001 (1.000–1.002)	0.07		
TBIL	1.007 (1.003–1.012)	0.002	1.001 (0.994–1.009)	0.763
DBIL	1.012 (1.005–1.018)	0.001	1.007 (0.999–1.016)	0.263
ALB	0.924 (0.884–0.966)	<0.001	0.932 (0.888–0.979)	0.005
PT	1.082 (0.985–1.188)	0.102		
FIB	1.22 (1.032–1.443)	0.02	1.121 (0.918–1.368)	0.263

## Data Availability

Data will be made available upon request.

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
