# Peer review of "Diagnostic and Prognostic Nomograms for Hepatocellular Carcinoma Based on PIVKA-II and Serum Biomarkers"

_diagnostics, 2023, doi:10.3390/diagnostics13081442_

Round 1

Reviewer 1 Report

The following exclusion criteria were not applied:

1) history of previous treatment (hepatic resection, liver transplant, trans-arterial chemo-embolization, radiofrequency, anti-angiogenetic drugs);

2) Child Pugh C;

3) obstructive jaundice;

4) estimated Creatinine Clearance <30 mL/min;

5) diagnosis of second extra-hepatic neoplasia;

6)metastasis.

In this study there is not application of Laboratory Standards Institute (CLSI) guidelines, They  are important to test serum samples from  healthy subjects to verify adherence to CLSI EP 28A3C; to evaluate normal range of values of PIVKA-II  and  AFP. 

Absence of the description of the pre-analytical phase and  preservation of the samples.

Reviewer 2 Report

Authors described a good markers diagnostic and prognostic combination to disciminate the HCC in case of HBV infection. Work is well written and conclusion are well supported by data submitted. Unfortunally patients sample size is no enohght big and in my opinion other confirm should be performed in this interesting way. 

However, I found a few minor revisions that authors should do:

- Authors should write the full name of AFP (Alpha Fetoprotein) one time in their manuscript.

- Authors forgot to definite HS (Healthy Samples) in the legend figure 1, however they definite in Material and Methods.

Fixed these little forgetfulnesses, I would like to suggest to accept the manuscript submitted.

Reviewer 3 Report

The mortality rate of HCC varies widely, depending on the stage of the disease and the treatment received. The overall 5-year survival rate for HCC is relatively low, ranging from 15% to 20%. However, if the cancer is detected and treated early, the 5-year survival rate can be much higher, up to 70% or more. This study aims at providing better guidance on therapeutic strategy and assessment of the prognosis of HCC by optimizing AFP with PIVKA-II-related diagnosis and prognosis model by combining with potential serum/plasma protein biomarkers in HCC.

While the study is certainly interesting, with solid data and analysis methods, an important point should be improved in the manuscript: While AFP is a widely used tumor marker for HCC, its sensitivity and specificity are limited and may not be elevated in all cases of HCC. Conversely, PIVKA-II is a more sensitive and specific marker for HCC, particularly in patients with low or normal AFP levels. PIVKA-II levels have been shown to increase with the progression of HCC and are associated with a poorer prognosis.

Thus, the combination of PIVKA-II and AFP measurements has improved the accuracy of HCC diagnosis and can help distinguish HCC from other liver diseases. However, it should be noted that PIVKA-II levels may also be elevated in patients with other types of cancer or liver diseases. Other tests may be needed to confirm the diagnosis of HCC.

This study is incremental compared to previously published manuscripts. The authors should better put their study into the perspective of previously published data and clearly emphasize the gain of their improved analysis for HCC prognosis.

Round 2

Reviewer 1 Report

Minor revision.

English needs to be revised by native speaker.

However the bibliography has not recent paper, but this new submitted version of the paper, I believe with a minimal revision on pubmed literature could be accepted. 

Author Response

Dear Reviewer 1,

Thank you for your precious time to constructive comments on our manuscript titled “Diagnostic and Prognostic Nomograms for Hepatocellular Carcinoma based on PIVKA-II and serum biomarkers” for Diagnostics (manuscript ID: diagnostics-2258595). We have revised the manuscript according to your suggestions and responded to the question raised as follows:

Point 1:

Quality of English Language

(x) Moderate English changes required

Response 1:

Thank you for the suggestion and we have invited a professional native English speaker to edit our manuscript.

Point 2:

Are all the cited references relevant to the research?  (x)Can be improved

Response 2:

We thank the reviewer for pointing this out and have updated the text with more recent literature.

Point 3:

English needs to be revised by native speaker.

Response 3:

Thank you very much for your suggestion. The language has been carefully corrected and modified by native English speaker.

Point 4:

However, the bibliography has not recent paper, but this new submitted version of the paper, I believe with a minimal revision on pubmed literature could be accepted.

Response 4:

Thank you for your valuable feedback regarding the bibliography in our manuscript. We appreciate your suggestion and are revising the manuscript to include more recent and relevant references.

For example, in the “Introduction” section, we have replaced the old references 2, 3, 6, 14 with the new references 2, 3, 7, 15, respectively. Please refer to our revised manuscript for other additions or deletions of cited literature.

We are grateful for your help in improving the quality of our work and look forward to submitting a revised version that meets your expectations.

Reviewer 3 Report

Thank you for quickly revising the manuscript. All concerns have been addressed.

Author Response

Thank you very much for taking the time to review our manuscript and for your valuable feedback.